# UNSUPERVISED DATA GENERATION FOR OFFLINE RE-INFORCEMENT LEARNING: A PERSPECTIVE FROM MODEL

## ABSTRACT

Offline reinforcement learning (RL) recently gains growing interests from RL researchers. However, the performance of offline RL suffers from the out-of-distribution problem, which can be corrected by environment feedback in online RL. Previous offline RL research focuses on restricting the offline algorithm in in-distribution even in-sample action sampling. In contrast, fewer work pays attention to the influence of the batch data. In this paper, we first build a bridge over the batch data and the performance of offline RL algorithms theoretically, from the perspective of model-based offline RL optimization. We draw a conclusion that, with mild assumptions, the distance between the state-action pair distribution generated by the behavioural policy, and the distribution generated by the optimal policy, accounts for the performance gap between the policy learned by model-based offline RL and the optimal policy. Secondly, we reveal that in task-agnostic settings, a series of policies trained by unsupervised RL can minimize the worst-case regret in the performance gap. Inspired by the theoretical conclusions, a framework named UDG (Unsupervised Data Generation) is composed to generate data and select proper data for offline training under tasks-agnostic settings. Empirical results on locomotive tasks demonstrate that UDG outperforms supervised data generation and previous unsupervised data generation in solving unknown tasks.

## 1 INTRODUCTION

Reinforcement learning (RL) recently gains significant advances in sequential decision making problems, with applications ranging from the game of Go (Silver et al., 2016; 2017), video games (Mnih et al., 2015; Hessel et al., 2018), to autonomous driving (Kiran et al., 2021) and robotic control (Zhao et al., 2020). However, the costly online trial-and-error process requires numerous samples of interactions with the environment which restricts RL from real world deployment. In the scenarios where online interaction is expensive or unsafe, we have to resort to offline experience (Levine et al., 2020). However, transplanting RL to offline can provoke disastrous error by falsely overestimating the out-of-distribution samples without correction from environment feedback. Despite recent advances on mitigating bootstrapped errors by constraining the policy in data distribution or even in data samples (Fujimoto & Gu, 2021), offline RL is still limited since they can barely generalize to out-of-distribution areas (Yu et al., 2020c). The inability of generalization of offline RL will be a serious issue when the batch data deviates from the optimal policy especially under the settings of multi-task, task transfer or task-agnostic. As plenty of research on online RL succeeds (Sodhani et al., 2021; Yu et al., 2020b;a; Laskin et al., 2021; Eysenbach et al., 2019; Sharma et al., 2020), we hope offline RL can cope with task-agnostic problems either. To this end, how the batch data distributes becomes the primal concern.

Recent research empirically shows that diversity in offline data improves performance on task transfer and solving multiple tasks (Lambert et al., 2022; Yarats et al., 2022). The diverse dataset is obtained from unsupervised RL by competitively training diverse policies (Eysenbach et al., 2019), or exploration emphasized pre-training (Liu & Abbeel, 2021a), and all of the generated data is fed to offline algorithms. However, these studies barely address the connection between the batch data

and the performance of offline RL theoretically. How the diversity of data contributes to solving task-agnostic problems remains unclear.

Our study addresses the connection between batch data and performance from a perspective of model-based offline optimization (Yu et al., 2020c). by examining the model prediction error and revealing the connection between the performance gap and the Wasserstein distance of the batch data distribution from the optimal distribution. We conclude that the offline trained policy will have higher return if the behavioural distribution is closer to the optimal distribution. We discover that, in task-agnostic scenarios, unsupervised RL methods which propel the policies far away from each other, approximately optimize the minimal regret to the optimal policy. Based on these theoretical analysis, we propose a framework

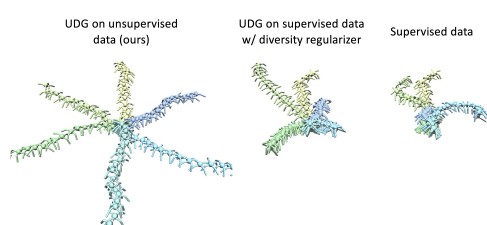

Figure 1: Rendered trajectories of offline trained policies in 6 Ant-Angle tasks. These tasks require the ant to move along 6 different directions. With diverse data buffers generated by unsupervisedly trained policies, our method UDG can solve all the tasks by offline reinforcement learning.

named unsupervised data generation (UDG) as illustrated in Figure 2. In UDG, a series of policies are trained with diversity rewards. They are used to generate batch data stored in different buffers. Before the offline training stage, the buffers are relabeled with given reward function corresponding to the task, and the buffer with highest return is sent to the offline algorithm.

The contributions in this work are three-fold. First, to our best knowledge, we are the first to establish a theoretical bond between the behavioural batch data and the performance of offline RL algorithms on *Lipschitiz continuous environments*. Second, we establish the criteria of minimal worst-case regret for data generation on task-agnostic problems. Third, we propose a new framework UDG for unsupervised offline RL and evaluate UDG on locomotive environments. Empirical results on locomotive tasks like Ant-Angle and Cheetah-Jump show that UDG outperforms conventional offline RL with random or supervised data.

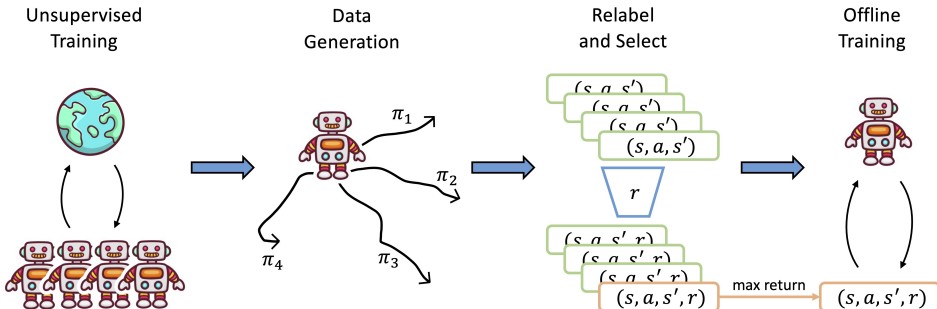

Figure 2: The framework of UDG. First a series of $K$ policies are trained simultaneously with diversity rewards. Second, collect rollout experience $(s, a, s')$ from each policy and construct a corresponding data buffer. Third, relabel the reward in the batch data with a designated reward function, and select the data buffer with the maximal average return. Finally train the agent on the chosen data by offline RL approaches.

## 2 RELATED WORK

**Offline RL**. In offline settings, vanilla online RL algorithms face the out-of-distribution problem that the Q function may output falsely high values on samples not in data distribution. To mitigate this issue, model-free offline RL methods cope with the out-of-distribution problem from two aspects, constraining the learned policy on the support of batch data (Fujimoto et al., 2019; Kumar et al., 2019; Wu et al., 2019; Peng et al., 2019; Siegel et al., 2020; Cheng et al., 2022; Rezaeifar et al., 2022; Zhang et al., 2022; Fujimoto & Gu, 2021), and suppressing Q values on out-of-distribution area (Agarwal et al., 2019; Kumar et al., 2020). To generalize beyond the batch data, model-based offline

RL methods employ transition models to produce extra samples for offline learning (Chen et al., 2021; Kidambi et al., 2020; Yu et al., 2020c; Matsushima et al., 2020). These methods can naturally generalize to areas where the model is accurate (Janner et al., 2019). Our theoretical work originates from MOPO (Yu et al., 2020c), a model-based offline algorithm by adding an uncertainty penalty to avoid unexpected exploitation when model is inaccurate. MOPO derives a performance lower bound w.r.t. model prediction error. In this paper, we focus on Lipschitz continuous environments which are common in locomotive tasks in MuJoCo (Todorov et al., 2012). By leveraging the Lipschitz geometry of transitions, we build a lower bound w.r.t. data distribution with weaker assumptions on data coverage, which is crucial in previous work (Wang et al., 2020; Chen & Jiang, 2019).

**Unsupervised RL**. Reinforcement learning heavily relies on the reward feedback of the environment for a specific tasks. However, recent research on unsupervised RL demonstrates training policies without extrinsic rewards enables the agent to adapt to general tasks (Laskin et al., 2021; Eysenbach et al., 2019). Without supervision from extrinsic rewards, unsupervised RL methods can either be driven by curiosity/novelty (Pathak et al., 2017; 2019; Burda et al., 2018), maximum coverage of the state space (Liu & Abbeel, 2021b; Campos et al., 2020; Yarats et al., 2021), and diversity of policies (Florensa et al., 2017; Lee et al., 2019; Liu & Abbeel, 2021a; Strouse et al., 2021; Kim et al., 2021). These methods all provide a pseudo reward derived from their own criteria. Our work employs a diverse series of policies to generate batch data for offline learning under task-agnostic settings. Therefore we utilize an unsupervised training paradigm in alignment with DIAYN (Eysenbach et al., 2019), DADS (Sharma et al., 2020), WURL (He et al., 2022), and choose WURL as base algorithm in accordance with our theoretical analysis.

**Offline dataset**. D4RL (Fu et al., 2020) and RL Unplugged (Gulcehre et al., 2020) are most commonly used offline RL benchmarks. The datasets in these benchmarks consist of replay buffers during training, rollout samples generated by a policy of a specific level, or samples mixed from different policies. Apart from benchmark datasets, exploratory data gains growing interest (Wang et al., 2022). Explore2Offline (Lambert et al., 2022) and ExORL (Yarats et al., 2022) both investigate into the role of batch data and construct a more diverse dataset with unsupervised RL algorithms for task generalization. In addition, experiments in ExORL show exploratory data can improve the performance of offline RL algorithms or even TD3 (Fujimoto et al., 2018). However, neither of these methods have theoretical analysis on the connection between data diversity and offline performance. Our work completes the picture of how diverse data improves offline RL performance. And we point out that data selection before offline training has considerable influence on performance, which is not addressed in previous work.

## 3 PRELIMINARIES

A Markov decision process (MDP) is formalized as $M = (\mathcal{S}, \mathcal{A}, T, r, p_0, \gamma)$, where $\mathcal{S}$ denotes the state space while $\mathcal{A}$ denotes the action space, $T(s'|s, a)$ the transition dynamics, $r(s, a)$ the reward function, $p_0$ the initial state distribution and $\gamma \in [0, 1)$ the discounted factor. RL aims to solve the MDP by finding a policy $\pi(a|s)$ maximizing the expected accumulated discounted return $\eta_M(\pi) := \mathbb{E}_{\pi,T,p_0}[\sum_{t=0}^{\infty} \gamma^t r(s_t, a_t)]$. The state value function $V_M^\pi(s_t) := \mathbb{E}_{\pi,T}[\sum_{k=0}^{\infty} \gamma^k r(s_{t+k}, a_{t+k})]$ provides an expectation of discounted future return from $s_t$ under policy $\pi$ and MDP $M$. Let $\mathbb{P}_{T,t}^\pi(s)$ be the probability of being in state $s$ at step $t$ when acting with policy $\pi$. The discounted occupancy measure of $\pi$ under dynamics $T$ is denoted by $\rho_T^\pi(s, a) := \frac{1}{c}\pi(a|s)\sum_{t=0}^{\infty} \gamma^t \mathbb{P}_{T,t}^\pi(s)$ where $c = 1/(1-\gamma)$ is the normalization constant. Likewise, $\rho_T^\pi(s) := \frac{1}{c}\sum_{t=0}^{\infty} \gamma^t \mathbb{P}_{T,t}^\pi(s)$ is the state occupancy distribution. The expected accumulated discounted return can be rewritten as $\eta_M^\pi = c\mathbb{E}_{\rho_T^\pi}[r(s, a)]$. $\hat{\rho}_T^\pi = \frac{1}{K}\sum_{i=1}^K \delta(s_i, a_i)$ denotes the empirical distribution of $\rho_T^\pi$ based on $K$ samples sampled from $\pi$ under transition dynamics $T$. $\delta(\cdot)$ denotes Dirac distribution.

Model-based RL approaches learn an estimated model $\hat{T}$ from interaction experience, which defines a model MDP $\hat{M} = (\mathcal{S}, \mathcal{A}, \hat{T}, r, p_0, \gamma)$. Similarly we have the expected return under the learned dynamics $\eta_{\hat{M}}(\pi) = c\mathbb{E}_{\rho_{\hat{T}}^\pi}[r(s, a)]$.

In offline settings, the RL algorithm optimizes the policy solely on a fixed dataset $\mathcal{D}_\beta = (s, a, r, s')$ generated by the behavioural policy $\pi^\beta$. $\pi^\beta$ can be one policy or a mixture of policies. Note that offline RL algorithms cannot interact with the environment or produce extra samples. In model-based offline RL, the algorithm first learn a transition model $\hat{T}$ from the batch data $\mathcal{D}_\beta$. At the

training stage, the algorithm executes $k$-step rollout using the estimated model from the state sample from $\mathcal{D}_\beta$. The generated data are added to another buffer $\mathcal{D}_m$. Both buffers are used in offline RL.

# 4 UDG: UNSUPERVISED DATA GENERATION

In order to find the connection between performance and data, we first review key propositions in MOPO (Yu et al., 2020c). As discussed before, offline RL faces a dilemma of out-of-distribution samples and lack of exploration. Model-based RL like MBPO (Janner et al., 2019) can naturally extend to the regions where the model predicts as well as the true dynamics. However, when the model is inaccurate, the algorithm may exploit the falsely high return regions, resulting in inferior test performance in true dynamics. MOPO first derives the performance lower bound represented by the model error, and then addresses the risk-return trade-off by incorporating the penalty represented by the error of the estimated dynamics into the reward of offline policy optimization.

We briefly summarize the derivation of the performance lower bound. First we introduce the telescoping lemma:

**Lemma 4.1.** *Let $M$ and $\hat{M}$ be two MDPs with the same reward function $r$, but different dynamics $T$ and $\hat{T}$ respectively. Denote $G_{\hat{M}}^\pi(s,a) := \mathbb{E}_{s' \sim \hat{T}(s,a)}[V_M^\pi(s')] - \mathbb{E}_{s' \sim T(s,a)}[V_M^\pi(s')]$. Then*

$$\eta_{\hat{M}}(\pi) - \eta_M(\pi) = c\gamma \mathbb{E}_{(s,a) \sim \rho_{\hat{T}}^\pi} \left[ G_{\hat{M}}^\pi(s,a) \right]. \tag{1}$$

If we have mild constraints on the value function $V_M^\pi \in \mathcal{F}$ where $\mathcal{F}$ is a bounded function class under a specific metric, then we can bound the gap $G_{\hat{M}}^\pi(s,a)$ with model error measured by corresponding integral probability measure (IPM) $d_\mathcal{F}$ (Müller, 1997),

$$|G_{\hat{M}}^\pi(s,a)| \leq \sup_{f \in \mathcal{F}} \left| \mathbb{E}_{s' \sim \hat{T}(s,a)}[f(s')] - \mathbb{E}_{s' \sim T(s,a)}[f(s')] \right| = d_\mathcal{F}(\hat{T}(s,a), T(s,a)). \tag{2}$$

Since we cannot access the true model $T$ in most cases, MOPO adopts an admissible error estimator $u : \mathcal{S} \times \mathcal{A} \to \mathbb{R}$ for $\hat{T}$, and have an assumption that for all $s \in \mathcal{S}.a \in \mathcal{A}$, $d_\mathcal{F}(\hat{T}(s,a), T(s,a)) \leq u(s,a)$. An uncertainty-penalized MDP $\tilde{M} = (\mathcal{S}, \mathcal{A}, \hat{T}, \tilde{r}, p_0, \gamma)$ is defined given the error estimator $u$, with the reward $\tilde{r}(s,a) := r(s,a) - \gamma u(s,a)$ penalized by model error.

By optimizing the policy in the uncertainty-penalized MDP $\tilde{M}$, MOPO has a following performance lower bound,

**Theorem 4.2** (MOPO). *Given $\hat{\pi} = \arg\max_\pi \eta_{\tilde{M}}(\pi)$ and $\epsilon_u(\pi) := c\mathbb{E}_{(s,a) \sim \rho_{\hat{T}}^\pi}[u(s,a)]$, the expected discounted return of $\hat{\pi}$ satisfies*

$$\eta_M(\hat{\pi}) \geq \sup_\pi \{\eta_M(\pi) - 2\gamma \epsilon_u(\pi)\}. \tag{3}$$

The theorem 4.2 reveals the optimality gap between $\pi^*$ and $\hat{\pi}$. Immediately we have $\eta_M(\hat{\pi}) \geq \eta_M(\pi^*) - 2\gamma\epsilon_u(\pi^*)$. This corollary indicates if the model error is small on the $(s,a)$ occupancy distribution under the optimal policy $\pi^*$ and dynamics $\hat{T}$, the optimality gap will be small. In order to find the deep connection between the batch data and the performance gap of model-based offline RL algorithms, in the following section, we directly analyze the model prediction deviation $d_\mathcal{F}(\hat{T}(s,a), T(s,a))$ instead of the error estimator $u(s,a)$.

## 4.1 THE CONNECTION BETWEEN BATCH DATA AND OFFLINE RL PERFORMANCE

Before presenting a lower bound of $\eta_M(\hat{\pi})$, here we make a few assumptions to simplify the proof. Some of these assumptions can be loosen and do not change the conclusion of the main result. The generalization of the theoretical analysis is discussed in Appendix C.

**Definition 4.3.** *Given a bounded subset $\mathcal{K}$ in the corresponding $d$-dimension Euclidean space $\mathbb{R}^d$. The diameter $B_\mathcal{K}$ of set $\mathcal{K}$ is defined as the minimum value of $B$ such that there exists $k_0 \in \mathbb{R}^d$, for all $k \in \mathcal{K}$, $\|k_0 - k\| \leq B$.*

**Assumption 4.4.** *The state space $\mathcal{S}$, the action space $\mathcal{A}$ are both bounded subsets of corresponding Euclidean spaces, with diameter $B_\mathcal{A} \ll B_\mathcal{S}$. The state transition function $T(s'|s,a)$ is deterministic and continuous.*

**Assumption 4.5.** *The state transition function $T(s'|s,a)$ is $L_T$-Lipschitz. For any $\pi$ the value function $V^\pi(s)$ is $L_r$-Lipschitz.*

As a consequence, given two state-action pairs $(s_1, a_1), (s_2, a_2)$, the next-state deviation under transition $T$ is upper bounded by

$$\|T(s_1, a_1) - T(s_2, a_2)\| \le L_T \|(s_1, a_1) - (s_2, a_2)\|. \tag{4}$$

**Assumption 4.6.** *The prediction model $\hat{T}(s'|s,a)$ is a non-parametric transition model, which means the model outputs the next state prediction by searching the nearest entry.*

Formally speaking, $\hat{T}$ has an episodic memory storing all input experience $\mathcal{D}_{\text{memory}} = \{(s_i, a_i, s'_i, r_i)\}_{i=1}^K$ (Pritzel et al., 2017). When feeding $\hat{T}$ a query $(s,a)$, the model returns $\hat{T}(s,a) = s'_k$ where $k = \arg\min_i \|(s,a) - (s_i, a_i)\|$. Assumption 4.6 implies $\hat{T}(s,a)$ is a deterministic function. Therefore combined with these two assumptions 4.4, 4.5, the gap $G_{\hat{M}}^\pi(s,a)$ defined in Lemma 4.1 is then bounded by

$$|G_{\hat{M}}^\pi(s,a)| \le L_r W_1(\hat{T}(s,a), T(s,a)) = L_r \|\hat{T}(s,a) - T(s,a)\|, \tag{5}$$

where $W_1$ is the 1-Wasserstein distance w.r.t. the Euclidean metric.

**Assumption 4.7.** *$\rho_T^{\pi^\beta}$ have a bounded support. The diameter of the support of distribution $\rho_T^{\pi^\beta}$ is denoted as $B_{\pi^\beta}$.*

Since the dataset size is out of our concern, we suppose the batch data is sufficient, such that $\hat{\rho}_T^{\pi^\beta} \approx \rho_T^{\pi^\beta} \approx \rho_{\hat{T}}^{\pi^\beta}$. For conciseness, we use $\rho_T^{\pi^\beta}$ in the following statements.

**Theorem 4.8.** *Given $\hat{\pi} = \arg\max_\pi \eta_{\tilde{M}}(\pi)$, the expected discounted return of $\hat{\pi}$ satisfies*

$$\begin{aligned}
\eta_M(\hat{\pi}) &\ge \eta_M(\pi^*) - 2c\gamma L_r L_T(W_1(\rho_T^{\pi^*}, \rho_{\hat{T}}^{\pi^*}) + W_1(\rho_T^{\pi^\beta}, \rho_T^{\pi^*})) \\
&\ge \eta_M(\pi^*) - 4c\gamma L_r L_T(W_1(\rho_T^{\pi^\beta}, \rho_T^{\pi^*}) + B_{\pi^\beta} + B_\mathcal{A}).
\end{aligned} \tag{6}$$

*If the batch data is collected from $N$ different policies $\pi_1^\beta, \ldots, \pi_N^\beta$, a tighter bound is obtained, where $\rho_T^{\pi^\beta}$ denotes the mixture of distribution $\rho_T^{\pi_i^\beta}, \ldots, \rho_T^{\pi_N^\beta}$,*

$$\eta_M(\hat{\pi}) \ge \eta_M(\pi^*) - 2c\gamma L_r L_T(W_1(\rho_T^{\pi^\beta}, \rho_T^{\pi^*}) + \min_i W_1(\rho_T^{\pi_i^\beta}, \rho_T^*) + 2B_{\pi^\beta} + 2B_\mathcal{A}). \tag{7}$$

The distance term $D_1 := W_1(\rho_T^{\pi^*}(s,a), \rho_{\hat{T}}^{\pi^*}(s,a))$ in the first line in Equation 6 is quite hard to estimate. However, we notice that the prediction model only outputs states in the episodic memory $\mathcal{D}_{\text{memory}}$ which implies $\text{supp}(\rho_{\hat{T}}^{\pi^*}(s)) \subseteq \text{supp}(\rho_T^{\pi^\beta}(s))$. We can naturally suppose that $\rho_{\hat{T}}^{\pi^*}(s,a)$ will not be too distinct from $\hat{\rho}_T^{\pi^\beta}(s,a)$. Therefore we can assume that $D_1 \approx D_2 := W_1(\rho_T^{\pi^\beta}(s,a), \rho_T^{\pi^*}(s,a))$, leading to an approximate lower bound free of $B_{\pi^\beta}$ and $B_\mathcal{A}$

$$\eta_M(\hat{\pi}) \ge \eta_M(\pi^*) - 4c\gamma L_r L_T(W_1(\rho_T^{\pi^\beta}, \rho_T^{\pi^*})). \tag{8}$$

For the data compounded by a mixture of policies, the approximate lower bound is

$$\eta_M(\hat{\pi}) \ge \eta_M(\pi^*) - 2c\gamma L_r L_T(W_1(\rho_T^{\pi^\beta}, \rho_T^{\pi^*}) + \min_i W_1(\rho_T^{\pi_i^\beta}, \rho_T^*)). \tag{9}$$

The detailed proof of Theorem 4.8 is presented in Appendix C. The main idea is to show the performance gap $\mathbb{E}_{(s,a) \sim \rho_{\hat{T}}^{\pi^*}} |G_{\tilde{M}}^{\pi^*}(s,a)|$ can be bounded by the distance between $\rho_{\hat{T}}^{\pi^*}$ and $\rho_{\hat{T}}^{\pi^\beta}$. The remaining part of proof utilizes triangle inequality to split the distance into two terms and then applies the assumptions to yield Equation 6.

**Interpretation:** Theorem 4.8 and Equation 8 suggest that the gap relies on $\pi^\beta$ and $\pi^*$. We denote the gap as $\mathcal{L}(\pi^\beta, \pi^*)$ such that $\eta_M(\hat{\pi}) \ge \eta_M(\pi^*) - \mathcal{L}(\pi^\beta, \pi^*)$. When the occupancy distribution of $\pi^\beta$ is closer to the occupancy distribution of the optimal policy $\pi^*$, the return of the policy optimized by MOPO will be closer to the optimal. Especially when $\pi^\beta = \pi^*$, MOPO can reach the optimal

return. Theorem 4.8 concentrates on the gap between $\pi^\beta$ and $\pi^*$. However, since the derivation does not involve the optimality of $\pi^*$, the inequality 6 holds true for any other policy $\pi$ instead of the optimal policy $\pi^*$. By substituting $\pi^*$ with $\pi^\beta$, we will obtain $\eta_M(\hat{\pi}) \geq \eta_M(\pi^\beta)$, which means the performance of the learned policy will perform no worse than the behavioral policy. This conclusion is consistent with the theoretical analysis in MOPO.

The second line of Equation 6 indicates a wider range of $\rho_T^{\pi^\beta}$ may enlarge the performance gap. This issue is mainly determined by the relation between $\rho_{\hat{T}}^{\pi^*}(s,a)$ and $\rho_T^{\pi^\beta}(s,a)$. In general cases, $\pi^*(a|s)$ will output actions that lead the next states closer to the optimal occupancy distribution. As a consequence, $\rho_{\hat{T}}^{\pi^*}(s,a)$ may be closer to $\rho_T^{\pi^*}(s,a)$ than $\rho_T^{\pi^\beta}(s,a)$. Therefore $D_1$ will be smaller than $D_2$. Nevertheless, $D_1 > D_2$ is still possible under some non-smooth dynamics or multi-modal situations. As a result, a broader distribution of $\rho_T^{\pi^\beta}(s,a)$ may impair MOPO performance.

## 4.2 THE MINIMAL WORST-CASE REGRET APPROACH

There are many cases where the optimal policy and the corresponding experience data is inaccessible for offline learning, e.g., (1) the reward function is unknown or partly unknown at the stage of data generation; (2) the batch data is prepared for multiple tasks with various reward functions; (3) training to optimal is expensive at the stage of data generation. Previous work in online RL suggests the diversity of policies plays the crucial role (Eysenbach et al., 2019). Especially in offline RL, where exploration is not feasible during training, the diversity of batch data should not be ignored.

Suppose we can train a series of $N$ policies simultaneously without any external reward. Our goal is to improve the diversity of the experience collected by policies $\{\pi_i\}_{i=1}^N$, such that there is at least one subset of the experience will be close enough to the optimal policy determined by the lately designated reward function at the offline training stage. Combined with Theorem 4.8, this objective can be formulated by

$$\min_{\pi_1,\dots,\pi_N \in \Pi} \max_{\pi^* \in \Pi} \min_i \mathcal{L}(\pi^i, \pi^*). \tag{10}$$

The inner $\min$ term $\text{REGRET}(\{\pi_i\}_{i=1}^N, \pi^*) := \min_i \mathcal{L}(\pi_i, \pi^*)$ represents the regret of the series of policies confronting the true reward function and its associate optimal policy. The $\max$ operator in the middle depicts the worst-case regret if any policy in the feasible policy set $\Pi$ has the possibility to be the optimal one. The outer $\min$ means the goal of optimizing $\{\pi_i\}_{i=1}^N$ is to minimize the worst-case regret. If the approximate lower bound is considered in Equation 8, the objective is equivalent to

$$\min_{\pi_1,\dots,\pi_N \in \Pi} \max_{\pi^* \in \Pi} \min_i W_1(\rho_T^{\pi_i}, \rho_T^{\pi^*}). \tag{11}$$

Directly optimizing a series of policies according to the minimax objective in Equation 11 inevitably requires adversarial training. Previous practice suggests an adversarial policy should be introduced to maximize $\text{REGRET}(\{\pi_i\}_{i=1}^N, \pi^*)$, playing the role of the unknown optimal policy. The adversarial manner of training brings us two main concerns. (1) Adversarial training may incur instability and require much more steps to converge (Arjovsky & Bottou, 2017; Arjovsky et al., 2017); (2) The regret only provides supervision signals to the policy nearest to $\pi^*$, which leads to low efficiency in optimization.

Eysenbach et al. (2021) proposed similar objective regarding unsupervised reinforcement learning. Under the assumptions of finite and discrete state space and, the quantity of policies $N$ should cover the number of distinct states in the state space $|\mathcal{S}|$, the minimal worst-case regret objective is equivalent to the maximal discriminability objective. Likewise, we propose a surrogate objective

$$\max_{\pi_1,\dots,\pi_N \in \Pi} \min_{i \neq j} W_1(\rho_T^{\pi_i}, \rho_T^{\pi_j}). \tag{12}$$

The surrogate objective shares the same spirit with WURL (He et al., 2022). Both of them encourage diversity of a series of policies w.r.t. Wasserstein distance in the probability space of state occupancy. Although the optimal solution of $\{\pi_i\}_{i=1}^N$ does not match the optimal solution in Equation 11 in general situations, both of them represent a kind of diversity. The relation between two objectives equals to the relation between finite covering and finite packing problems, which are notoriously difficult to analyze even in low-dimension, convex settings (Böröczky Jr et al., 2004; Toth et al., 2017). Nevertheless, we assume the gap will be small and the surrogate objective will be a satisfactory proxy of Equation 11 as previous literature does in the application of computational graphics (Schlömer et al., 2011; Chen & Xu, 2004). Refer to Appendix D for more details.

### 4.3 PRACTICAL IMPLEMENTATION

To achieve diversity, practical algorithms assign a pseudo reward $\tilde{r}_i$ to policy $\pi_i$. The pseudo reward usually indicates the "novelty" of the current policy w.r.t. all other policies. Similar to WURL, we adopt pseudo reward $\tilde{r}_i := \min_{j \neq i} W_1(\rho_T^{\pi_j}, \rho_T^{\pi_i})$ which is the minimum distance from all other policies. We compute the Wasserstein distance using amortized primal form estimation in consistent with WURL (He et al., 2022).

In semi-supervised cases, only part of reward function is known. For example, in Mujoco simulation environments in OpenAI Gym (Brockman et al., 2016), the complete reward function is composed of a reward related to the task, and general rewards related to agent's health, control cost, safety constraint, etc. We can train the series of policies with a partial reward and a pseudo reward simultaneously by reweighting two rewards with a hyperparameter $\lambda$. Moreover, the complete reward and the diversity-induced pseudo reward can be combined to train a diverse series of policies for generalization purposes.

The policies are trained with Soft Actor-Critic method (Haarnoja et al., 2018). The network model of the actors are stored to generate experience $\mathcal{D}_1, \ldots, \mathcal{D}_K$ for offline RL. When a different reward function is used at the offline training stage. The reward will be relabeled with $r(s, a)$. At the offline learning stage, we choose the best buffer and feed it to MOPO. The overall algorithm is illustrated in Figure 2 and formally described in Algorithm 1.

---

**Algorithm 1** Unsupervised data generation for offline RL in task-agnostic settings

---

**Require:** $K$ policies $\pi_1, \ldots, \pi_K$. $K$ empty buffers $\mathcal{D}_1, \ldots, \mathcal{D}_K = \{\}$. Maximum buffer size $N$.

1: Train $\pi_i, i = 1, \ldots, K$ with SAC w.r.t. diversity rewards $\tilde{r}_i := \min_{j \neq i} W_1(\rho_T^{\pi_j}, \rho_T^{\pi_i})$.
2: Let each $\pi_i$ interacts with environment for $N$ steps and fill $\mathcal{D}_i$ with transitions $(s, a, s')$.
3: Acquire the task and relabel all transitions in $\mathcal{D}_1, \ldots, \mathcal{D}_K$ with given $r(s, a)$.
4: Evaluate each buffer and calculate the average return $\bar{G}_i, i = 1, \ldots, K$.
5: Select the buffer $\mathcal{D}_k$ where $k = \arg\max_i \bar{G}_i$
6: Train the policy by MOPO with $\mathcal{D}_k$.

---

## 5 EXPERIMENTS

Based on our framework of UDG in Figure 2, we conduct experiments on two locomotive environments requiring the agent to solve a series of tasks with different reward functions at the offline stage. Both of the tasks are re-designed Mujoco environments. Ant-Angle is a task modified from Ant environment. In Ant-Angle, the agent should actuate the ant to move from the initial position to a specific direction on the x-y plane. The agent is rewarded by the inner product of the moving direction and the desired direction. The goal is to construct a dataset while the desired direction is unknown until the offline stage. Cheetah-Jump is another task for evaluation, modified from HalfCheetah environment. The reward in Cheetah-Jump consists of three parts, control cost, velocity reward, and jumping reward. At the data generation stage, the agent can only have access to the control cost and the velocity reward for reducing energy cost of actuators and moving the cheetah forward. The jumping reward is added in offline training, by calculating the positive offset of the cheetah on the z axis. Likewise, a crawling reward can be added to encourage the cheetah to lower the body while moving forward.

Our experiments mainly focus on two aspects: (1) How does UDG framework perform on the two challenging tasks, Ant-Angle and Cheetah-Jump? (2) Can experimental results match the findings in the theoretical analysis?

### 5.1 EVALUATION ON TASK-AGNOSTIC SETTINGS

To answer question (1), we construct three types of data buffer. The first is generated by unsupervisedly trained policies with the objective in Equation 12. The second is created by one supervisedly trained policy that maximizes a specific task reward. The third is the combination of two, which means the polices are trained with both task reward and diversity reward, reweighted by $\lambda$. We call

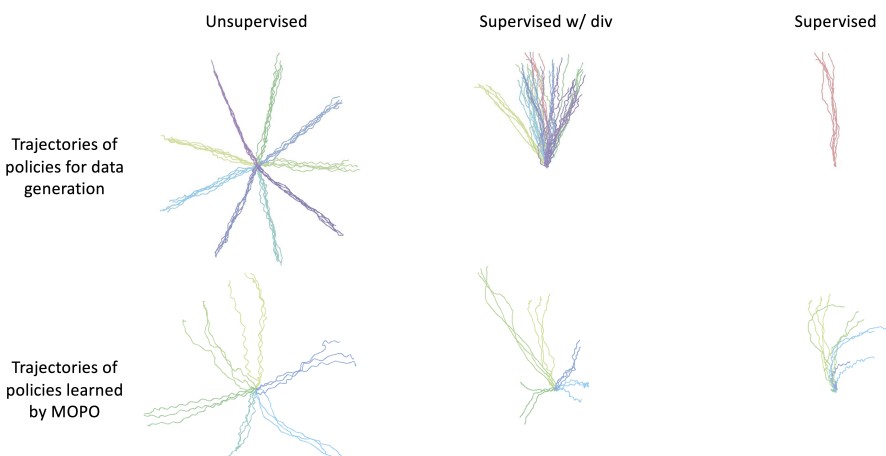

Figure 4: The trajectories of ant on the x-y plane. The lower left figure demonstrates UDG can solve all 6 offline tasks on account of the diversity of unsupervised learned polices. The trajectories in the lower right figure shows the policy learned by MOPO cannot generalize to the directions that largely deviate from $0°$.

the combination as supervised training with diversity regularizer. Note that the supervised method contains one data buffer. Another two methods have a series of 10 buffers w.r.t. 10 polices and only one buffer is selected during offline training.

We evaluate three kinds of data buffers on Ant-Angle. The supervised policy and the supervised policies with diversity regularizer are provided with reward to move in direction $0°$, the upper direction in Figure 4. The diversity reward is calculated on ant position instead of the whole state space, considering that the dimension of state space is extremely high. We evaluate three approaches on 6 offline tasks of moving along the directions of $0°$, $60°$, $120°$, $180°$, $240°$ and $300°$. As Figure 4 shows, the policies trained with unsupervised RL are evenly distributed over the x-y plane. Therefore in the downstream offline tasks, no matter what direction the task needs, there exists at least one policy that could obtain relatively high reward. The trajectories of policies trained by MOPO confirm that UDG can handle all 6 tasks. Meanwhile the policy trained to move in the direction $0°$ generates narrow data and MOPO cannot perform well on other directions. The policies trained with combined reward have wider range of data distribution. Especially, an policy deviates to circa $30°$ and consequently the policy trained by MOPO acquires high reward in the $60°$ task.

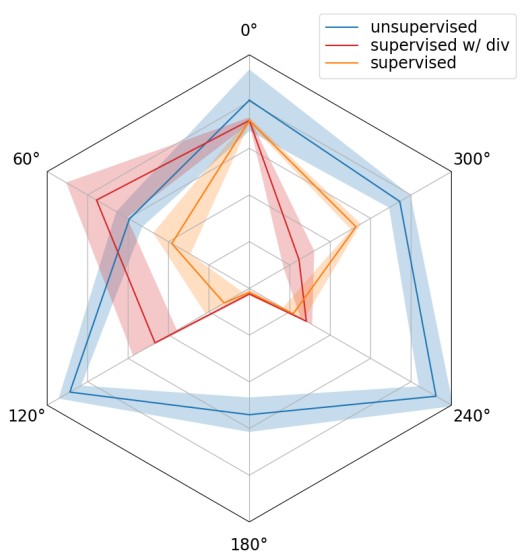

Figure 3: Results on Ant-Angle tasks. The data buffers of all three methods are evaluated by MOPO with 3 random seeds. The darker lines represent the average evaluation return. The lighter areas depict the standard deviation of the return.

| Task | $c_z$ | random | diverse |
|---|---|---|---|
| Cheetah-Jump | 15 | 1152.98±120 | **1721.25±56** |
| Cheetah-Crawl | -15 | 1239.00±57 | **1348.19±274** |

Table 1: Returns on two offline tasks Cheetah-Jump and Cheetah-Crawl. Two datasets consist of 5 policies trained with base rewards and base rewards plus diversity rewards respectively.

In Cheetah-Jump tasks, we relabel the data by adding an extra reward $c_z(z - z_0)$ where $z_0$ is the initial position on the z axis, and $c_z$ is the coefficient of the extra reward. $c_z$ can either be positive or negative. For positive values of $c_z$, the cheetah is encouraged to jump while running forward. For negative values, crawling on the floor receives higher reward. We train 5 polices each for base rewards and base rewards plus diversity rewards, denoted by "random" and "diverse" respectively.

## 5.2 EFFECTS OF THE RANGE OF DATA DISTRIBUTION

| Angle | top 1 | top 2 mixed | all mixed |
|---|---|---|---|
| 0° | 1236.26±247 | **1437.24±31** | 989.13±65 |
| 60° | 910.70±121 | **1285.31±66** | 593.88±434 |
| 120° | **1362.46±104** | 917.10±218 | 281.88±301 |
| 180° | 829.65±139 | **1034.41±224** | 717.68±120 |
| 240° | **1416.80±160** | 1373.72±73 | 850.82±62 |
| 300° | **1141.68±100** | 1087.26±137 | 817.77±37 |

Table 2: Returns on Ant-Angle tasks with different angles trained on different datasets. Top 1 dataset is the data buffer with highest return. Top 2 mixed dataset is a mixture of two highest-rewarded buffers. All mixed dataset is a mixture of all data buffers generated by unsupervisedly trained policies.

With the help of Ant-Angle environment and the policies learned by unsupervised RL, we conduct several experiments to verify the conclusions from theoretical derivations. Apart from the data buffer with maximum return, we build a data buffer denoted by "all mixed", by mixing data generated by all 10 policies. We also mix the data from top 2 polices to create a "top 2 mixed" buffer.

Referring to the upper left figure in Figure 4, the "top 2 mixed" data buffer includes two policies lying on the left and the right side near direction $0°$. the top two distributions have similar distance from the optimal distribution. Therefore the mixed distribution $\rho_T^{\pi^\beta}$ has similar distance to the optimal compared with the nearest distribution $W_1(\rho_T^{\pi^\beta}, \rho_T^{\pi^*}) \approx \min_i W_1(\rho_T^{\pi_i^\beta}, \rho_T^*)$. However, when all policies are mixed, it is obvious $W_1(\rho_T^{\pi^\beta}, \rho_T^{\pi^*}) > \min_i W_1(\rho_T^{\pi_i^\beta}, \rho_T^*)$. According to Equation 9, the top 2 mixed dataset will get higher return than all mixed dataset. Table 2 and results in Appendix F have verified this claim. From another aspect, the top 2 mixed data buffer has a wider distribution than the top 1 buffer. Therefore the top 2 mixed buffer has a larger radius $B_{\pi^\beta}$ which may worsen performance according to Equation 6. Surprisingly, the top 2 mixed buffer makes higher return than top 1 single buffer. We can conjecture that $B_{\pi^\beta}$ plays an insignificant role in the lower bound and the approximation in Equation 8 and 9 is proper. In addition, the wide spread of the mixed data may improve the generalization ability of the transition model in MOPO, which contributes to the higher return than top 1 data buffer.

## 6 CONCLUSION

In this study we propose a framework UDG addressing data generation issues in offline reinforcement learning. In order to solve unknown tasks at the offline training stage, UDG first employs unsupervised RL and obtains a series of diverse policies for data generation. The experience generated by each policy is relabeled according to the reward function adopted before the offline training stage. The final step is selecting the data buffer with highest average return and feeding the data to model-based offline RL algorithms like MOPO. We provide theoretical analysis on the performance gap between the offline learned policy and the optimal policy w.r.t the distribution of the batch data. We also reveal that UDG is an approximate minimal worst-case regret approach under the task-agnostic setting. Our experiments evaluate UDG on two locomotive tasks, Ant-Angle and Cheetah-Jump. Empirical results on multiple offline tasks demonstrate UDG is overall better than data generated by a policy dedicated to solve a specific task. Additional experiments show that the range of data distribution has minor effects on performance and the distance from the optimal policy is the most important factor. It is also confirmed that choosing the data buffer with highest return is necessary for better performance compared to other unsupervised data generation approaches.

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

## A  LIMITATIONS

Our derivation is based on the continuous state space with assumption that the transition function and the value functions are Lipschitz. There are some tasks may break the assumptions, i.e., pixel based tasks like Atari, non-smooth reward functions in goal reaching tasks Tassa et al. (2018). Therefore, it is necessary to verify the feasibility of UDG on these tasks in future deployment. We also adopt a non-parametric transition model in derivation. In practical model-based offline RL approaches, neural models have greater generalization ability than the non-parametric model. The influence of data distribution on neural models is not addressed by this work. In addition, whether can UDG be generalized with model-free offline RL algorithms remains unclear. Another limitation is at the unsupervised training stage, the diversity reward is calculated on the low dimensional space where the reward function is defined, i.e., the x-y plane in Ant-Angle. This requires prior knowledge of how the reward is computed. Nevertheless, the limitations mentioned above indicate interesting further research.

## B  SOCIETAL IMPACT

The UDG framework contains a stage of unsupervised RL. At this stage, the agent is not provided with any reward for solving any task. During the process of training, the agent may unexpectedly exploit the states in unsafe regions. Especially when deployed in realistic environments, it could incur damage of the environment or the robotic agent itself, or cause injury in robot-human interactions. Any deployment of UDG in the real world should be carefully designed to avoid safety incidents.

## C    PROOFS AND ADDITIONAL THEORETICAL ANALYSIS

### C.1    COMPLETE PROOF OF THEOREM 4.8

First we introduce the telescoping lemma in MOPO:

**Lemma C.1.** *Let $M$ and $\hat{M}$ be two MDPs with the same reward function $r$, but different dynamics $T$ and $\hat{T}$ respectively. Denote $G_{\hat{M}}^{\pi}(s, a) := \mathbb{E}_{s' \sim \hat{T}(s,a)}[V_M^{\pi}(s')] - \mathbb{E}_{s' \sim T(s,a)}[V_M^{\pi}(s')]$. Then*

$$\eta_{\hat{M}}(\pi) - \eta_M(\pi) = c\gamma \mathbb{E}_{(s,a) \sim \rho_{\hat{T}}^{\pi}} \left[ G_{\hat{M}}^{\pi}(s, a) \right]. \tag{13}$$

To give a proof of the main theorem, we make these assumptions:

**Definition C.2.** *Given a bounded subset $\mathcal{K}$ in the corresponding $d$-dimension Euclidean space $\mathbb{R}^d$. The diameter $B_{\mathcal{K}}$ of set $\mathcal{K}$ is defined as the minimum value of $B$ such that there exists $k_0 \in \mathbb{R}^d$, for all $k \in \mathcal{K}$, $\|k_0 - k\| \le B$.*

**Assumption C.3.** *The state space $\mathcal{S}$, the action space $\mathcal{A}$ are both bounded subsets of corresponding Euclidean spaces, with diameter $B_{\mathcal{A}} \ll B_{\mathcal{S}}$. The state transition function $T(s'|s, a)$ is deterministic and continuous.*

**Assumption C.4.** *The state transition function $T(s'|s, a)$ is $L_T$-Lipschitz. For any $\pi$ the value function $V^{\pi}(s)$ is $L_r$-Lipschitz.*

**Assumption C.5.** *The prediction model $\hat{T}(s'|s, a)$ is a non-parametric transition model, which means the model outputs the next state prediction by searching the nearest entry.*

Formally speaking, $\hat{T}$ has an episodic memory storing all input experience $\mathcal{D}_{\text{memory}} = \{(s_i, a_i, s_i', r_i)\}_{i=1}^N$. When feeding $\hat{T}$ a query $(s, a)$, the model returns $\hat{T}(s, a) = s_k'$ where $k = \arg\min_i \|(s, a) - (s_i, a_i)\|$.

**Assumption C.6.** *$\rho_T^{\pi^{\beta}}$ have a bounded support. The diameter of the support of distribution $\rho_T^{\pi^{\beta}}$ is denoted as $B_{\pi^{\beta}}$.*

Since the dataset size is out of our concern, we suppose the batch data is sufficient, such that $\hat{\rho}_T^{\pi^{\beta}} \approx \rho_T^{\pi^{\beta}} \approx \rho_{\hat{T}}^{\pi^{\beta}}$. For conciseness, we use $\rho_T^{\pi^{\beta}}$ in the proof.

First we review the definition of Wasserstein distance from the view of optimal transport and introduce a lemma used in the proof.

**Definition C.7** ($W_1$ distance). *Two discrete distributions $p(x) = \frac{1}{N} \sum_{i=1}^N \delta(x_i)$ and $q(y) = \frac{1}{M} \sum_{j=1}^M \delta(y_j)$ are defined on an Euclidean space $\mathcal{X}$. The Wasserstein distance between $p$ and $q$ is defined as*

$$W_1(p, q) = \min_{\gamma \in \Gamma(N,M)} \sum_{i=1}^N \sum_{j=1}^M \gamma_{ij} \|x_i - y_j\|, \tag{14}$$

*where $\Gamma(N, M)$ is the class of all transporting matrices $\gamma$ satisfying (1) $\forall i, j, 0 \le \gamma_{ij} \le 1$, (2) $\forall i$, $\sum_{j=1}^M \gamma_{ij} = \frac{1}{N}$ and (3) $\forall j, \sum_{i=1}^N \gamma_{ij} = \frac{1}{M}$. The optimal matrix is denoted as $\gamma^*$.*

$\gamma$ can be explain as a joint distribution on $\mathcal{X} \times \mathcal{X}$ with marginal distribution $p(x)$ and $q(y)$ respectively.

**Definition C.8** (Nearest transporting cost). *The nearest transporting cost from $p$ to $q$ is defined as*

$$V_1(p \to q) = \min_{\gamma \in \Gamma(N,M)} \sum_{i=1}^N \sum_{j=1}^M \gamma_{ij} \|x_i - y_j\|, \tag{15}$$

*where $\Gamma(N, M)$ is the class of all transporting matrices $\gamma$ satisfying (1) $\forall i, j, 0 \le \gamma_{ij} \le 1$ and (2) $\forall i, \sum_{j=1}^M \gamma_{ij} = \frac{1}{N}$.*

Since $V_1$ loosen the constraints in the optimization problem, it is obvious $V_1 \le W_1$. The definition of Wasserstein distance and nearest transporting cost can naturally generalize to continuous distribution $p(x)$.

**Lemma C.9** (Triangle inequality). *For discrete distributions $p(x), q(y), r(z)$ on $\mathcal{X}$,*

$$V_1(p \to q) \leq W_1(p, r) + V_1(r \to q). \tag{16}$$

*Proof.* The RHS of 16 is the total cost of transporting mass from $p$ to $r$ and from $r$ to $q$. The inequality holds true by the optimality of the matching matrix in $V_1(p \to q)$ and the triangle inequality in the Euclidean space. □

**Theorem C.10.** *Given $\hat{\pi} = \arg\max_\pi \eta_{\tilde{M}}(\pi)$, the expected discounted return of $\hat{\pi}$ satisfies*

$$\begin{aligned} \eta_M(\hat{\pi}) &\geq \eta_M(\pi^*) - 2c\gamma L_r L_T(W_1(\rho_T^{\pi^*}, \rho_{\hat{T}}^{\pi^*}) + W_1(\rho_T^{\pi^\beta}, \rho_T^{\pi^*})) \\ &\geq \eta_M(\pi^*) - 4c\gamma L_r L_T(W_1(\rho_T^{\pi^\beta}, \rho_T^{\pi^*}) + B_{\pi^\beta} + B_{\mathcal{A}}). \end{aligned} \tag{17}$$

*If the batch data is collected from $N$ different policies $\pi_1^\beta, \ldots, \pi_N^\beta$, a tighter bound is obtained, where $\rho_T^{\pi^\beta}$ denotes the mixture of distribution $\rho_T^{\pi_1^\beta}, \ldots, \rho_T^{\pi_N^\beta}$,*

$$\eta_M(\hat{\pi}) \geq \eta_M(\pi^*) - 2c\gamma L_r L_T(W_1(\rho_T^{\pi^\beta}, \rho_T^{\pi^*}) + \min_i W_1(\rho_T^{\pi_i^\beta}, \rho_T^*) + 2B_{\pi^\beta} + 2B_{\mathcal{A}}). \tag{18}$$

*Proof.* Assumption C.5 implies $\hat{T}(s, a)$ is a deterministic function. Therefore combined with these two assumptions C.3, C.4, the gap $G_{\hat{M}}^\pi(s, a)$ defined in Lemma C.1 is then bounded by

$$\begin{aligned} |G_{\hat{M}}^\pi(s, a)| &\leq \sup_{f \in \mathcal{F}} \left| \mathbb{E}_{s' \sim \hat{T}(s,a)}[f(s')] - \mathbb{E}_{s' \sim T(s,a)}[f(s')] \right| \\ &= L_r W_1(\hat{T}(s, a), T(s, a)) = L_r \|\hat{T}(s, a) - T(s, a)\|, \end{aligned} \tag{19}$$

where $W_1$ is the 1-Wasserstein distance w.r.t. the Euclidean metric. Assumption C.5 tells us, given any $(s, a)$ pair, the non-parametric model's output is $\hat{T}(s, a) = T(\hat{s}, \hat{a})$ where $(\hat{s}, \hat{a}) \in \mathcal{D}_{\text{memory}}$ is the nearest point to $(s, a)$. Note that $\rho_T^{\pi^\beta} = \frac{1}{K} \sum_{i=1}^K \delta(s_i, a_i)$ is the uniform distribution on $\mathcal{D}_{\text{memory}}$. With assumption C.4 combined, we have

$$|G_{\hat{M}}^\pi(s, a)| \leq L_r \|\hat{T}(s, a) - T(s, a)\| = L_r \|T(\hat{s}, \hat{a}) - T(s, a)\| \leq L_r L_T \|(\hat{s}, \hat{a}) - (s, a)\|. \tag{20}$$

By definition of the nearest transporting cost and Lemma 16, we have

$$\begin{aligned} \mathbb{E}_{(s,a) \sim \rho_{\hat{T}}^{\pi^*}} |G_{\hat{M}}^{\pi^*}(s, a)| &\leq L_r L_T V_1(\rho_{\hat{T}}^{\pi^*} \to \rho_T^{\pi^\beta}) \leq L_r L_T(W_1(\rho_{\hat{T}}^{\pi^*}, \rho_T^{\pi^*}) + V_1(\rho_T^{\pi^*} \to \rho_T^{\pi^\beta})) \\ &\leq L_r L_T(W_1(\rho_{\hat{T}}^{\pi^*}, \rho_T^{\pi^*}) + W_1(\rho_T^{\pi^*}, \rho_T^{\pi^\beta})) \end{aligned} \tag{21}$$

Considering $\rho_{\hat{T}}^{\pi^*}(s)$ is strictly on the support of $\rho_T^{\pi^\beta}(s)$, we can bound the first term by

$$W_1(\rho_{\hat{T}}^{\pi^*}, \rho_T^{\pi^*}) \leq W_1(\rho_T^{\pi^*}, \rho_T^{\pi^\beta}) + 2B_{\pi^\beta} + 2B_{\mathcal{A}} \tag{22}$$

If $\rho_T^{\pi^\beta}$ is a mixture of $N$ behaviour policies, by the definition of $V_1$ cost, we have

$$\begin{aligned} \mathbb{E}_{(s,a) \sim \rho_{\hat{T}}^{\pi^*}} |G_{\hat{M}}^{\pi^*}(s, a)| &\leq L_r L_T(W_1(\rho_{\hat{T}}^{\pi^*}, \rho_T^{\pi^*}) + V_1(\rho_T^{\pi^*} \to \rho_T^{\pi^\beta})) \\ &\leq L_r L_T(W_1(\rho_{\hat{T}}^{\pi^*}, \rho_T^{\pi^*}) + \min_i W_1(\rho_T^{\pi^*}, \rho_T^{\pi_i^\beta}))). \end{aligned} \tag{23}$$

Substitute $\pi$ in MOPO main theorem with $\pi^*$ and we have

$$\eta_M(\hat{\pi}) \geq \eta_M(\pi^*) - 2c\gamma \mathbb{E}_{(s,a) \sim \rho_{\hat{T}}^{\pi^*}} |G_{\hat{M}}^{\pi^*}(s, a)|, \tag{24}$$

which completes the proof. □

## C.2 GENERALIZATION OF THE PROOF

Here we try to analyze how sensitive is our proof to the assumptions.

First we look into assumption C.3. If the true dynamics $T(s, a)$ and the estimated transition model $\hat{T}(s, a)$ are no longer deterministic, we will be unable to reduce $W_1(\hat{T}(s, a), T(s, a))$ to $\|\hat{T}(s, a) - T(s, a)\|$. Actually, under stochastic dynamics, we can assume that the Lipschitz condition of $T$ implies

$$\forall (s, a), (s', a'), \frac{W_1(T(s, a), T(s', a'))}{\|(s, a) - (s', a')\|} \leq L_T. \tag{25}$$

Additionally we suppose $\hat{T}$ randomly outputs one of $M$ results out of all experience $(s, a, s'_1), \ldots, (s, a, s'_M)$ with the same $(s, a)$. Therefore we have

$$\begin{aligned} W_1(\hat{T}(s, a) - T(s, a)) &\leq W_1(\hat{T}(\hat{s}, \hat{a}) - T(\hat{s}, \hat{a})) + W_1(T(\hat{s}, \hat{a}) - T(s, a)) \\ &\leq W_1(\hat{T}(\hat{s}, \hat{a}) - T(\hat{s}, \hat{a})) + L_T \|(\hat{s}, \hat{a}) - (s, a)\|. \end{aligned} \tag{26}$$

The second term corresponds to the term in Equation 20. And the first term is the sample error, which will diminish to zero if the number of samples goes to $\infty$. Another concern is, under stochastic dynamics, there is *almost zero probability* to sample the same $(s, a)$ pair. In order to reduce estimation error, we should adopt the same manner in Neural Episodic Control (Pritzel et al., 2017) to search nearest $K$ entries in memory and output a weighted $s'$. In conclusion, under stochastic dynamics, Equation 20 will have an extra term of sample error. The error will be small if sample size is sufficiently large.

In practical implementation like MOPO, neural networks are adopted to estimate transition dynamics. It is expected that neural transition models will have the ability to generalize to areas out of the support of data distribution than episodic memories. As a consequence, $W_1(\rho_T^{\pi^\beta}, \rho_T^{\pi^*})$ in Equation 17 becomes small if $\hat{T}$ is accurate under the optimal policy $\pi^*$. In this situation, the lower bound suggests the performance of offline RL will be higher. Nevertheless, our work pays more attention on the influence of data than model generalization. Therefore we leave the concrete analysis of neural models as future work.

## D RELATION TO COVERING AND PACKING PROBLEMS

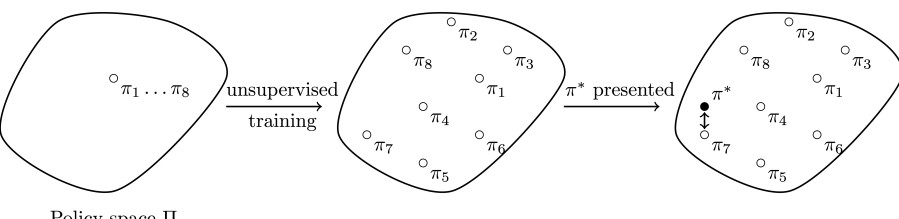

Figure 5: Illustration of the minimal worst-case regret approach. At the unsupervised training stage, we hope to push $N$ policies far away from each other such that any policy (the unknown optimal policy) will not be too far away from the nearest policy.

In Section 4.2, we propose a minimal worst-case approach of finding $N$ policies that the maximum regret is minimal. The corresponding objective is

$$\min_{\pi_1, \ldots, \pi_N \in \Pi} \max_{\pi^* \in \Pi} \min_i W_1(\rho_T^{\pi_i}, \rho_T^{\pi^*}). \tag{27}$$

Our proposed objective is maximizing the minimal distance of every two policies in $N$ policies.

$$\max_{\pi_1, \ldots, \pi_N \in \Pi} \min_{i \neq j} W_1(\rho_T^{\pi_i}, \rho_T^{\pi_j}). \tag{28}$$

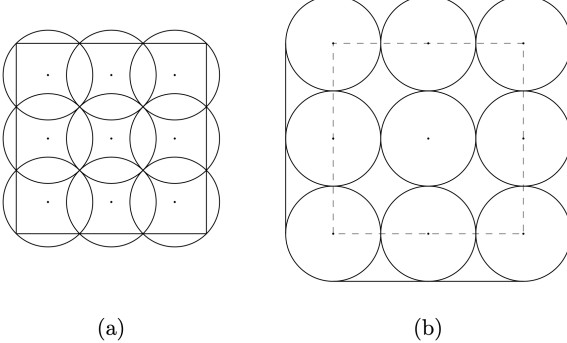

(a)                                            (b)

Figure 6: Illustration of covering and packing problems on a 3x3 square. (a) The covering problem is equivalent to the optimization problem in Equation 31. For any position in the square, the nearest distance from 9 points is no more than the radius. (b) The packing problem is equivalent to the optimization problem in Equation 33, since we do not have the border constraint in objective 28. The nearest distance between every two points is no less than the diameter.

We claim that the objective in Equation 27 is equivalent to a space covering problem and the objective in Equation 28 is equivalent to a space packing problem. Here we formulate the covering and packing problems defined in (Toth et al., 2017).

For given convex sets $K$ and $C$, and a positive integer $N$, the two quantities are corresponding to a packing problem and a covering problem respectively:

$$M_p(K, C, N) = \inf\{\lambda | N \text{ congruent copies of } C \text{ can be packed in } \lambda K\}, \tag{29}$$

and

$$M_c(K, C, N) = \sup\{\lambda | N \text{ congruent copies of } C \text{ can cover } \lambda K\}. \tag{30}$$

If we assume $C = B^2$ is the unit disk in $E^2$ (2-dimensional Euclidean space) and $K$ is an arbitrary convex set in $E^2$. We consider two quantities from optimization problems:

$$G_N = \min_{x_1,\ldots,x_N \in K} \max_{x \in K} \min_i \|x - x_i\|, \tag{31}$$

and

$$H_N = \max_{x_1,\ldots,x_N \in K} \min_{i \neq j} \|x_i - x_j\|. \tag{32}$$

We define an alternative packing quantity

$$\tilde{M}_p(K, C, N) = \inf\{\lambda | N \text{ congruent copies of } C \text{ can be packed in } \lambda K + C\}, \tag{33}$$

where $\lambda K + C$ means the Minkowski sum. It is obvious that we have $G_N = 1/M_c(K, B^2, N)$ and $H_N = 2/\tilde{M}_p(K, B^2, N)$. Figure 6 provides an illustration of the two problems. Although the solutions of packing and covering problems are generally not the same, they both encourage some kind of even distribution of $N$ points in a given space. Therefore, the two optimization objectives are good proxy objectives of each other in practical implementations. For example, (Schlömer et al., 2011) proposes an algorithm of generating $N$ points in a square under the *Poisson-disk* criterion which demands that no two points are closer than a certain minimal distance. The Poisson criterion is equal to the optimization objective in Equation 32. However in the proposed "farthest-point optimization" method, the iterative process selects a position farthest from every point and places a new point at that position. This process can approximately optimize the objective in Equation 31. That is the reason why we use the maximum min-dist criterion to minimize the worst-case regret.

## E    IMPLEMENTATION DETAILS

### E.1    POLICIES FOR DATA GENERATION

The policies used in training are Soft-Actor-Critic policies (Haarnoja et al., 2018). We adopt SAC with double Q networks and one Gaussian actor. Both of the critic and actor have 3 hidden layers.

| Hyperparamter | Value |
|---|---|
| $\gamma$ | 0.99 |
| $\tau$ | 0.005 |
| target update interval | 1 |
| updates per step | 1 |
| batch size | 256 |
| startup steps | 10000 |
| total steps | 200000 |
| lr for critic | 0.0003 |
| lr for actor | 0.0003 |
| $\alpha$ | 0.2 |
| auto $\alpha$ tuning | False |
| hidden dim | 256 |
| replay size | 200000 |

Table 3: Hyperparameters in policy training.

The training hyperparameters are listed in Table 3. In unsupervised training, we use 10 policies in Ant-Angle and 5 policies in Cheetah-Jump. The first policy is training with no diversity reward. Other policies are trained combined with the diversity reward proposed in WURL (He et al., 2022). The policies are changed in order by every 1 episode in training. Denote the original reward for policy $i$ (zero reward, task reward or base reward) as $r_i$ and the diversity reward as $d_i$. The composite reward is reweighted with $\lambda$, as $\tilde{r}_i = r_i + \lambda d_i$. We use $\lambda = 1$ in Ant-Angle and $\lambda = 10$ in Cheetah-Jump.

In the data generation process, we sample 1 million transitions per policy, with maximum episode length 200. Therefore each data buffer has 5k episodes of transitions. The mixed data buffers are constructed with equal number of transitions per policy. The mixed buffer also has 1 million transitions. We cut off the dimensions in Ant-Angle to 27, in accordance with MBPO (Janner et al., 2019), since the original 111 dimensions is quite hard for downstream model training. The rewards in data buffers are relabeled before offline training.

### E.2 OFFLINE TRAINING

The backbone algorithm at the offline training stage is MOPO (Yu et al., 2020c). The hyperparameters are the same with the official repo. The tunable hyperparameters are rollout length $h$ and penalty coefficient $\lambda$. In each experiment, we choose the best $(h, \lambda)$ pair with $h$ in range of $\{1, 2, 5\}$ and $\lambda$ in range of $\{1, 5, 7, 10\}$. The model learning has maximum 100 iterations and the policy training has maximum 1000 iterations. As the evaluation performance is quite unstable in MOPO, we report the best evaluation score in each task. The evaluation score is averaged over 10 test episodes.

### E.3 COMPUTATION RESOURCE

Experiments in our work are deployed on a server with AMD EPYC 7H12 64-Core Processor and 8 NVIDIA GeForce RTX 3090 GPUs. Training at the data generation stage requires approximate 6hrs per million-step per GPU. For example, training 10 policies in Ant-Angle with each policy 200k steps requires 12hrs on one GPU, and training 5 policies in HalfCheetah with each policy 1M steps requires 30hrs on one GPU. Training with MOPO has various time consumption depending on dataset size, rollout length $h$ and environment. A typical experiment of Ant-Angle requires 8hrs per GPU. And an experiment of HalfCheetah requires 12hrs per GPU. Considering random seeds, hyperparameter tuning, our work costs about 10k GPU hours.

# F ADDITIONAL RESULTS

## F.1 ABLATION

To demonstrate the data generated by UDG is superior to data generated by other unsupervised policies. We adopt DIAYN as our baseline (Eysenbach et al., 2019). The diversity reward is added into the composite reward in the same manner . We train DIAYN with 10 policies in Ant-Angle and set $\lambda = 1$. Results are reported in Table 4.

| Environment | buffer mean | random | buffer mean | diayn |
|---|---|---|---|---|
| Ant-Angle 0° | -3.1 | 33.28±3.98 | 80.83 | 401.64±33.80 |
| Ant-Angle 60° | -13.21 | 26.63±8.06 | 88.11 | 322.59±101.09 |
| Ant-Angle 120° | -10.12 | 66.72±80.35 | 140.59 | 98.75±43.92 |
| Ant-Angle 180° | 3.1 | 30.61±13.77 | 95.27 | 719.91±57.03 |
| Ant-Angle 240° | 13.21 | 64.12±89.93 | 93.33 | 522.62±194.93 |
| Ant-Angle 300° | 10.12 | 42.10±34.53 | 125.36 | 535.15±60.42 |

Table 4: Offline training returns on random policy and unsupervised policies trained by DIAYN. Policies trained by DIAYN cannot reach comparable performance with WURL. Results on the data generated by DIAYN are also inferior to WURL generated data.

## F.2 FULL RESULTS

We also try UDG on conventional tasks like HalfCheetah. We train 5 policies with diversity reward and use snapshots at different steps to generate transitions. We mix the data from all policies and compare it to a single policy and a mixture of random initialized policies. Results are shown in Figure 7. The mixture of 5 policies has minor improvements in offline performance.

Table 3-6 show additional results in Ant-Anagle tasks with buffer mean return presented.

| Environment | unsup. (top 1) | sup. w/ reg. (top 1) | sup. |
|---|---|---|---|
| Ant-Angle 0° | **1236.26±247.24** | 1103.14±24.20 | 1103.14±24.20 |
| Ant-Angle 60° | 910.7±121.40 | **1160.01±282.32** | 588.71±172.55 |
| Ant-Angle 120° | **1362.46±104.37** | 716.05±204.08 | 191.73±172.82 |
| Ant-Angle 180° | **829.65±138.69** | 36.68±11.93 | 25.65±21.83 |
| Ant-Angle 240° | **1416.80±159.50** | 432.58±54.25 | 332.94±95.79 |
| Ant-Angle 300° | **1141.68±99.69** | 376.88±141.84 | 809.46±45.28 |

Table 5: Returns on Ant-Angle tasks trained on different datasets. The datasets are in accordance with those in Figure 3 in the main paper.

| Environment | top 1 | top 2 mixed | all mixed | top and last mixed |
|---|---|---|---|---|
| Ant-Angle 0° | 1236.26±247.24 | **1437.24±31.17** | 989.13±64.71 | 1008.16±167.70 |
| Ant-Angle 60° | 910.7±121.40 | **1285.31±65.86** | 593.88±434.34 | 765.88±129.22 |
| Ant-Angle 120° | **1362.46±104.37** | 917.10±218.36 | 281.88±301.17 | 707.83±166.97 |
| Ant-Angle 180° | 829.65±138.69 | **1034.41±223.68** | 717.68±120.35 | 816.51±33.46 |
| Ant-Angle 240° | **1416.80±159.50** | 1373.72±72.60 | 850.82±62.35 | 946.28±87.33 |
| Ant-Angle 300° | **1141.68±99.69** | 1087.26±136.89 | 817.77±36.50 | 1056.75±37.48 |

Table 6: Returns on Ant-Angle tasks trained on different datasets. This table is the extended results of Table 2 in the main paper.

| Environment | top 1 | sup.w/ reg | sup. | top 2 mixed | all mixed | top and last |
|---|---|---|---|---|---|---|
| Ant-Angle 0° | 1888.01 | 1839.20 | 1839.20 | 1796.57 | -19.92 | 81.68 |
| Ant-Angle 60° | 1623.52 | 1709.94 | 1118.49 | 1561.78 | -75.85 | -129.43 |
| Ant-Angle 120° | 1835.61 | 300.87 | -720.71 | 1673.06 | -55.93 | 29.16 |
| Ant-Angle 180° | 1723.55 | -1365.33 | -1839.20 | 1698.92 | 19.92 | -82.78 |
| Ant-Angle 240° | 1883.29 | -189.63 | -1118.49 | 1723.12 | 75.85 | 130.33 |
| Ant-Angle 300° | 1777.88 | 1388.05 | 720.71 | 1658.29 | 55.93 | -28.57 |

Table 7: Buffer mean returns on Ant-Angle tasks of different datasets. Top 1, top 2 mixed, all mixed and top and last datasets are all selected from the data buffers generated by unsupervisedly trained policies. Sup. w/ reg. denotes the supervisedly trained policies with diversity regularizer.

| Environment | $c_z$ | random | diverse |
|---|---|---|---|
| Cheetah-Jump | 15 | 325.73 | 1536.62 |
| Cheetah-Crawl | -15 | 792.21 | 1592.05 |

Table 8: Buffer mean returns on two offline tasks Cheetah-Jump and Cheetah-Crawl. Two datasets consist of 5 policies trained with base rewards and base rewards plus diversity rewards respectively.

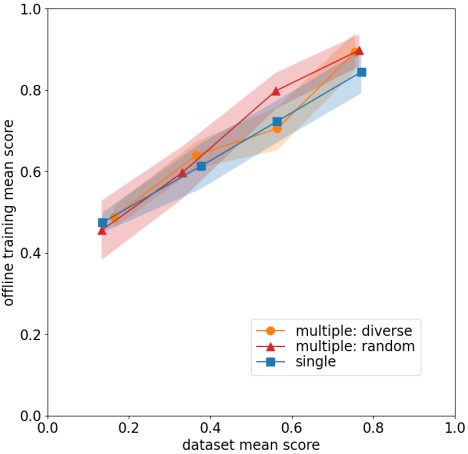

Figure 7: Diverse data on HalfCheetah tasks. Returns are shown in normalized scores according to D4RL. Each point represents the mean score of the behaviour policy used and the mean score of policy learned by MOPO.

