# OpenReview forum: "Unsupervised Data Generation for Offline Reinforcement Learning: A Perspective from Model"
_ICLR.cc/2024/Conference — ICLR 2024 Conference Withdrawn Submission_

### Official Review · Reviewer_FQFn · 2023-10-24

**Soundness:** 1 poor
**Presentation:** 2 fair
**Contribution:** 2 fair
**Rating:** 3
**Confidence:** 4

**Summary:**

This paper proposes unsupervised data generation for offline reinforcement learning in the task agnostic setting. It first establishes a theoretical bond between the data generated by the behavior policy and the performance of offline RL algorithms, from the model-based perspective. To exploit the theoretical bond, the paper proses a minimax approach to generate offline RL data in an unsupervised fashion. Experiments on cheetah and ant angle environments show the effectiveness of the proposed approach.

**Strengths:**

The topic of generating data for offline RL in an unsupervised fashion seems appealing.

The paper provides a theoretical analysis of the problem of unsupervised data generation and instantiates its theoretical findings as a practical minimax algorithms. Empirical experiments show the effectiveness of unsupervised data generation.

**Weaknesses:**

The evaluation is limited. The experiments are only carried out on two environments: cheetah and ant angle, while even mujoco alone has over 10 tasks where the authors can validate their method. For example, environments from https://arxiv.org/pdf/2110.15191.pdf may be adopted.

Comparison with prior literature seems inadequate. As discussed in the "Unsupervised RL" section of related works, there is extensive literature for unsupervised RL driven by "curiosity, maximum coverage of states, and diversity of policies". The paper claims that "these methods all provide a pseudo reward derived from their own criteria". But why can't those methods be run to collect diverse data for offline RL?

The paper seems to set its theoretical analysis as one of its major contributions but a few key steps of the theoretical analysis are not explained well. For example, the bound in theorem 4.8 seems to have a constant dependence on $B_{\pi^\beta}$ and$B_{\mathcal{A}}$. This can make the bound arbitrarily bad. Furthermore, the distance term $D_1 = W_1(\rho^{\pi^\*}_T(s,a), \rho^{\pi^\*}_\hat{T}(s,a))$ seems to be arbitrarily bad because there is no guarantee of the performance of the transition model $\hat{T}$ under a different distribution.

**Questions:**

Please see weakness.

---

### Official Review · Reviewer_jP6G · 2023-10-31

**Soundness:** 2 fair
**Presentation:** 3 good
**Contribution:** 2 fair
**Rating:** 3
**Confidence:** 3

**Summary:**

Offline Reinforcement Learning (Offline RL) is plagued by the out-of-distribution problem. This study investigates the relationship between batch data and performance from the perspective of model-based offline optimization. It does so by examining model prediction errors and highlighting the connection between performance disparity and the Wasserstein distance between the batch data distribution and the optimal distribution. The conclusion drawn from this research is that if the behavior distribution is closer to the optimal distribution, the offline-trained policy will achieve a higher return. The authors introduce a framework named UDG (Unsupervised Data Generation) to generate data and select appropriate data for offline training in task-agnostic settings. Empirical results on locomotion tasks show that UDG surpasses supervised data generation and previous unsupervised data generation methods in addressing unknown tasks.

**Strengths:**

According to the author's claim, this paper is the first work to establish a theoretical link between behavior batch data and the performance of offline RL algorithms in Lipschitz continuous environments.

The authors introduce a framework, UDG, for unsupervised offline RL. Experimental results show that on robotic locomotion tasks like Ant-Angle and Cheetah-Jump, UDG outperforms traditional offline RL that uses either random or supervised data.

**Weaknesses:**

The motivation of this article is insufficient. One fundamental notion we perceive in offline reinforcement learning is the high cost associated with data collection. However, the paper assumes that data can be freely collected in a virtual environment, which seems contradictory to the original intention behind the design of offline reinforcement learning. This indicates a potential misalignment in perspective. Furthermore, the core idea presented in the study appears to lack originality, raising concerns about its novelty.

The experimental validations are not exhaustive enough. The number of environments tested appears limited, and there is a noticeable lack of substantial theoretical backing to solidify the claims. Moreover, the subsequent offline algorithms employed are solely based on MOPO. It would have added significant value if the paper had benchmarked the performance across other offline algorithms as well, to present a more holistic understanding.

The proof conclusions in this paper are not sufficiently novel. The proof regarding the distance between the strategy employed for data collection and the optimal strategy has been similarly discussed in section 3.1 of the article "Is Pessimism Provably Efficient for Offline RL". That article had previously drawn a conclusion similar to the findings of this work. The distinction is that the authors of this paper establish it under the certain Lipschitz MOPO condition, but the final proof conclusion still resembles that of this study.

**Questions:**

5.1 In the experimental section, how is the algorithm for the supervised part designed? Please show more details on the comparison of data generation algorithms.

---

### Official Review · Reviewer_sJsv · 2023-11-04

**Soundness:** 1 poor
**Presentation:** 2 fair
**Contribution:** 1 poor
**Rating:** 3
**Confidence:** 4

**Summary:**

This paper casts a theoretical light on the question of data optimality when performing offline RL. Concretely, it lays out a theoretical framework to understanding what sort of data results in optimal performance under offline RL algorithms, and then uses these results to derive an offline data collection scheme to gather such data. They show that data gathered under this unsupervised RL scheme (UDG) performs better than other approaches.

**Strengths:**

* Paper is on a relevant topic to Offline RL, which is understanding the importance of the data itself on the performance of algorithms.

**Weaknesses:**

The paper has a significant number of shortcomings.

* The paper has a number of language issues. For instance, the title itself is grammatically incorrect; the subtitle should be something like "A model-based perspective" or "A perspective from model-based Reinforcement Learning". In the abstract, "Previous offline RL research focuses on restricting the offline algorithm in in-distribution even in-sample action sampling. In contrast, fewer work pays attention to the influence of the batch data" -> "Previous offline RL research has focused on restricting the offline algorithm to lie in-distribution, such as preventing out-of-distribution action sampling. In contrast, there has been less focus on the influence of the batch data itself." As another point, it claims that TD3+BC constrains the action sampling to be in-distribution, which is clearly not true. TD3+BC regularizes action selection towards in-distribution samples, it doesn't explicitly constrain the actions selected. I recommend the authors carefully revise their manuscript.
* The theory was not easy to follow, and I had several issues with it. For instance, much of the beginning of the theory is simply a copy of MOPO; this facsimile should have been relegated to the appendix. Indeed, up to equation 5, the assumptions and proofs (e.g., having the gap measure, Lipschitzness in the value functions) are exactly that from MOPO. The paper then makes a number of assumptions that don't seem to have much bearing on reality. For instance, they claim the prediction model will only output states that are in the input memory. But we know from abundant model-based deep RL literature, such as [1] that models can often output states very far away from the actual provided datasets. Similarly, assuming the batch is sufficient such that the empirical and model-based state-action distributions are roughly equivalent seems very strong (made after 4.7). Finally, the proof and main "intuition" seems a bit tautological: if we could attain a dataset close to the optimal (potentially unknown) trajectory distribution under $\pi^*$, then we can get better performance. I feel like such a finding is somewhat obvious (e.g., if we had something like the expert dataset, we should be expected to perform better), and doesn't need theory in the first place. Finally, the UDG algorithm seems like a clone of WURL? I'm not too sure where the main differences are, and its justification is somewhat handwavey (e.g., we need better coverage, so here's a good coverage algorithm).
* Empirical results aren't that strong. Results only shown on 3 domains (Cheetah Jump/Crawl and Ant Angles), and moreover, no statistical testing is done to show that their approach improves over the baselines. For instance, their approach is bolded for Cheetah Crawl, but the random algorithm seems to perform roughly as well as UDG.
* The presented data-collection algorithm (e.g., deriving policies that aim to be as different to previous policies) seems very similar to [2], whereby a population of $N$ policies are learned such that they behave as differently as possible to all prior trained policies. The work also presents a theoretical proof showing improvement guarantees given this diverse data collection scheme which the authors should discuss. The authors should also cite and discuss [3], who also show that diverse data can improve offline RL, but also observe that not all diversity is useful, and can harm performance (thus they recommend only sharing when a performance improvement can be achieved); I feel this result is the also main result of this work? Again, there is theory presented in this paper that the authors would do well to consider and compare to their own.

[1] Revisiting Design Choices in Offline Model-Based RL, Lu et al., ICLR 2022
[2]  Learning General World Models in a Handful of Reward-Free Deployments, Xu et al., NeurIPS2022
[3]  Conservative Data Sharing for Multi-Task Offline Reinforcement Learning, Yu et al., NeurIPS2021

**Questions:**

* Could the authors improve the writing and technical content of their manuscript? I've pointed some issues, but there are more in the manuscript.
* Could the authors clarify the novelty and necessity of their analysis and theory?
* Could the authors provide statistical testing on their empirical results?
* Could the authors cite the additional work shown, and explain the relevance of their work compared to them?

---

### Official Review · Reviewer_DbRP · 2023-11-07

**Soundness:** 2 fair
**Presentation:** 2 fair
**Contribution:** 2 fair
**Rating:** 3
**Confidence:** 3

**Summary:**

This paper aims to address the out-of-distribution challenge that arises in batch RL. The paper first studies the connection between the optimality gap and the distance between state-action distributions generated by optimal policy and sampling policy, respectively. Then, the authors discover an unsupervised RL method that minimizes the worst-case regret. Inspired by this theory, the authors propose UDG, a
Finally, the authors conduct extensive experiments on several locomotive tasks to show that UDG performs well in real-world RL challenges.

**Strengths:**

1. While the observation that optimality gap depends on the distance between optimal policy and sampling policy is somewhat well-known in the line of offline RL research (e.g., [1][2][3], to name a few), the authors draw an explicit connection between the Wasserstein distance of visitation measures and the optimality gap, which is novel and interesting to me.


[1] Y. Duan, Z. Jia, M. Wang, Minimax-Optimal Off-Policy Evaluation with Linear Function Approximation. (2020)

[2] T. Xie et al., Policy Finetuning: Bridging Sample-Efficient Offline and Online Reinforcement Learning. (2022)

[3] W. Zhan, et al., Provable Offline Preference-Based Reinforcement Learning. (2023)

**Weaknesses:**

1. The approximation in theory parts need more careful handling. In particular, in page 5 of the paper, the authors assume that (1) the empirical visitation $\hat rho ^\pi_T ~ \rho^\pi_T ~ \rho^\pi_{\hat T}$. In addition, the authors further assume that (2) the visitation $\rho^{\pi^*}_{\hat T} ~ \hat \rho^{\pi^\beta}_{T}$. I have a few questions regarding the two hypothesis.

* While I agree that (1) hold if dataset is sufficiently large, I wonder what would be the challenge of offline RL in such a rich data setting, where theoretically any online RL algorithms are going to work decently?
* Given that (1) is correct, can we infer from (2) that \rho^{\pi^*}\_{\hat T} is similar to \rho^{\pi^\beta}\_{\hat T} and \rho^{\pi^*}\_{T} is similar to \rho^{\pi^\beta}\_{T} ? In that case, the optimality gap between \pi^\beta and \pi^* is small (upper bounded by r_max * |\rho^{\pi^*}\_{T} -  \rho^{\pi^\beta}\_{T}|_1), which would make imitation learning suffice as a vanilla option to offline RL.

2. The paper needs a revised abstract and introduction to correctly reflect the problem that it is trying to resolve. Though the introduction and abstract suggest that the authors are aiming to resolve the offline RL challenge, the algorithm proposed, UDG, is actually an self-supervised RL algorithm, where (different from the typical offline RL setup) agent is allowed to interact with the environment. While the proposed UDG falls in a self-supervised RL paradigm, I assume that the highlight that authors try to show is the offline data selection part, where the authors follow their theoretical finding to optimize offline dataset to train. If offline RL is indeed the challenge that authors are trying to resolve, the paper would be better presented if authors highlight how the UDG framework and the theoretical findings can help tacking the offline RL challenge or evaluating the offline dataset given in a typical offline RL setting.

3. (Minor) Though it may not be the focus of this work, it would be better to inclued some baseline comparison (e.g., the WURL that the authors proposed), which would make it easier for audience to place this work in this line of research.

**Questions:**

See Weaknesses.